# Increasing the Circularity of Packaging along Pharmaceuticals Value Chain

Hanna Salmenperä [1,*], Sari Kauppi [2], Helena Dahlbo [1] and Päivi Fjäder [1]

1 Centre for Sustainable Consumption and Production, Finnish Environment Institute, 00790 Helsinki, Finland; helena.dahlbo@syke.fi (H.D.); paivi.fjader@syke.fi (P.F.)
2 Programme for Sustainable Circular Economy, Finnish Environment Institute, 00790 Helsinki, Finland; sari.kauppi@syke.fi
* Correspondence: hanna.salmenpera@syke.fi

**Abstract:** Pharmaceutical packaging is a complex group of products, the main purpose of which is to protect the medicine and forward information. Pharmaceutical packaging waste is generated and accumulated along the various phases and practices of the value chain. In general, the amount of packaging has been growing during the increasing political pressure to reduce waste and to increase the circulation of materials. The goals and solutions are expected to be found in the circular economy; however, the literature on circular pharmaceutical packaging is lacking. This study explores the key factors when promoting the circularity of pharmaceutical packaging along its value chain. This was conducted by reviewing the legislation, elaborating the value chain and analysing the data from focus group discussions with stakeholders. The results show that various barriers, such as legislation, a lack of information or interaction between stakeholders, but also rigid practices, block product design for circularity. In the developing circularity of packaging, the causal links along the value chain must be understood. Chemical recycling technologies are expected to resolve the challenges of maintaining clean cycles. Further studies are needed to demonstrate the environmental benefits of increasing circularity along the value chain of pharmaceutical packaging.

**Keywords:** pharmaceutical packaging; packaging waste; circular economy; barriers; value chain; stakeholders

## 1. Introduction

Pharmaceutical packaging (PhP) is a complex group of products consisting of various materials, including different plastic polymers, glass, paper, steel, cardboard and metals. There are numerous requirements for PhP materials depending on the needs, but securing the quality and function of pharmaceutical products is of utmost importance. Requirements include protection from external impacts such as humidity, oxygen, light exposure, changes in temperature, biological contamination and mechanical damage. Regulatory requirements, such as identification by labelling and product information, are also needed for safe usage and to avoid the marketing of counterfeit medicines, as well as to ensure the correct information is provided to users [1]. Lastly, the convenience of use is also a required attribute [2]. Therefore, it could be stated that medicine is not a medicine without proper packaging.

The use of packaging and packing materials has increased substantially over the years. This has led to a growing volume of packaging waste and challenges in waste management, e.g., [3,4]. Like many other packaging, it is also typical of PhP that they are disposable. All investments and materials used for packaging are lost if they cannot be recycled. Packaging is one of the key product value chains for which urgent actions are needed to change unsustainable practices. The EU has a strong focus on packaging, which is defined to be one of the sectors where the potential to circularity is high and the targets and actions for recycling and recyclability for packages are ambitious [3]. Such actions include a reduction

in packaging and packaging waste, the increased use of reusable and recyclable designs and a reduction in the use of complex packaging materials [3]. Although the circular economy (CE) has been proposed as an approach to solving the problems arising from the wasteful and linear use of natural resources, the current western economies are still far from circular [5]. CE barriers have become a major subject of academic research [6–10]. Furthermore, there seems to be no commonly adopted definition for the circular economy, e.g., [5,11,12]. We defined that CE is an economic system where materials and products are used in a way that maintains their value in the material cycle as long as possible, while energy is used efficiently and waste is avoided and recycled. Geissdoerfer et al. [13] also emphasizes the importance of the connection between CE and sustainability. In our study, we follow the principles of CE [11,12,14] applicable in the case of PhP, i.e., optimizing products and practices to avoid waste, dematerialization, design for recyclability, reuse and recycling, keeping in mind that the transition to a CE requires a systemic change. Studies on the circularity of packaging have been performed in recent years, e.g., on possibilities to change collection systems, to change the materials used and increase the separation efficiency, e.g., [15,16]. Yet, the literature and discussion concerning the possibilities to increase the circularity of PhP are lacking.

Earlier studies show that well-designed packaging can lead to significant environmental benefits as losses along the value chain decrease [17,18]. Additionally, the environmental benefits gained from recycling packaging often prove to be more numerous than benefits from energy recovery. However, environmental benefits depend greatly on various factors, beginning from the material in question. PhPs can be complex materials because of their many requirements, and the recycling technology needed is still in an early stage [19]. Due to the biohazard contamination risk, healthcare plastics may require some sorting before recycling, whereas cardboard, aluminium and glass are best managed within their own recycling streams [20]. The challenge is that promoting circularity leads to trade-offs between environmental impacts and residues of harmful substances in packaging [21]. However, innovations in PhP are possible [22,23]. Pareek and Khunteta [23] define various trends that currently affect development, including aspects such as counterfeit prevention or child-resistant packaging, but also eco-friendly packaging. They state that the pressure to develop sustainable packaging has even begun to affect PhP [23]. Although the material selection depends most importantly on the character of the pharmaceutical, we claim that the possibilities to increase the circularity of packaging, i.e., material choices, as well as the reduction in packaging, should be studied and developed further.

The value chain can be defined as the range of activities which are required to bring a product or service from an initial idea and product design, through the different phases of production (logistics, transforming inputs and packaging), to delivery to final users and disposal after use [24]. The packaging materials that flow along the value chain of pharmaceutical products are not yet well known [25]. The journey of pharmaceuticals from the production site to the consumer is long and complicated. Along the way, pharmaceuticals are stored and distributed by several actors. The practices, along with the value chain, greatly affect the amount of packaging waste produced. Therefore, there is a need to identify the key value chain activities to gain an insight into the possibilities of increasing the circularity of PhP. Further, the value chain is constantly under governance exercised by parties inside the value chain such as companies or by parties outside it such as EU. According to Kaplinsky [24], the role of governance can vary from legislative (e.g., environmental standards for suppliers) to judicial (e.g., monitoring conformance to the standards) or executive (e.g., assisting supplier to meet the standards). While the various modes of governance dictate the efficient functioning of the value chain [24], the value chain management to promote circularity is lacking research [26,27]. Additionally, the role packaging plays in supply chain efficiency seems to be neglected [28].

This study contributes to the literature on promoting the CE in the sector of pharmaceuticals. It increases the understanding of the possibilities of operators along the PhP value chain, but also the key elements for changing and adopting novel practices. This

information facilitates the parties inside and outside the value chain, both the practitioners and the legislators, in their future work in introducing innovative measures and solutions. As this study pays additional attention to plastic PhP, it also contributes to the global debate on reducing the environmental impacts of plastics. The main research question of this study asks what kind of factors need to be taken into consideration when promoting the circularity of PhPs along their value chain.

## 2. Materials and Methods

### 2.1. PhP Materials and Range

This study focused on PhP that covers primary, secondary and tertiary packaging for tablets and capsules. Further attention was also paid to plastic packaging. Packaging for other formats of pharmaceuticals, e.g., ampoules or vials, was not included in the study. However, the range and purpose of various packaging needs to be elaborated to understand the big picture of developing the circularity of packaging.

PhPs are usually divided into either primary or secondary and tertiary packaging depending on the contact with the actual pharmaceutical product [23]. Primary packaging materials are in contact with the pharmaceutical product, but secondary packaging may be needed, for example, to ensure proper labelling. Tertiary packing such as plastic wrappings around pallets may be added to facilitate the transportation of products between value chain operators.

PhP consists of various materials such as cardboard (e.g., boxes), paper (e.g., labels and leaflets), glass (e.g., ampoules, vials and bottles), plastics (e.g., closures, bottles, blisters, bags, tubes and laminates with paper or foil), metals (e.g., foils) and rubber (e.g., closures) [1]. Usually, PhP is also a combination of different materials. However, plastics represent one major material type in primary PhP [2,23]. The commonly used plastic polymers in primary PhP include polyvinylchloride (PVC), polyethylene (PE), polypropylene (PP), polyethyleneterephtalate (PET), polycarbonate (PC), polyamide (PA), polystyrene (PS) and polyurethane (PU). Various blisters and bottles are common because almost half of all pharmaceuticals are administered orally. According to Holopainen [25], almost half (48%) of medicines sold in Finland are packed in blisters. At the same time, the share of bottles is approximately 10% [25]. The plastics in blisters are mainly PVC, but also PP, PS and/or PE [19] are used, while bottles and containers are usually composed of PE, PP, PET and glass.

### 2.2. The Methodology

This qualitative study on systemic challenge built an overall picture of the key aspects along the value chain to improve the circularity of PhP. A Finnish PhP chain was 'the case study' of this research. The packaging industry faces similar challenges in Europe compared to the world. The purpose of a case study is to provide empirical in-depth understanding about a contemporary phenomenon in a real-life context [29].

The answer to our research question could best be found by using diverse research data and combining different approaches, which is the recommended method of data collection in a case study [29]. Incorporating legislation review as well as value chain description enable systemic observations from circularity of PhP along the value chain and avoid drawing detached conclusions from the whole. Therefore, methodology of this study included three phases (Figure 1). The process started with discussions with key stakeholder experts to create an understanding about the overall view on the operational environment and existing knowledge gaps in developing PhP circularity. The phases 1a and 1b formed the solid basis for the analysis in phases 2 and 3. Firstly, we reviewed the EU legislative framework (Figure 2). This review focused on regulations on PhP; principles, materials, packaging types and altering the PhPs. We also looked at legislation on waste management principles, packaging waste, plastic waste and producer responsibility. Two experts were consulted to ensure that all relevant regulations were included in the review. Subsequently, we sketched a picture of the value chain with the help of operators working

with pharmaceuticals packaging. This included identifying the focal points where PhP was changed or added and also where waste was generated (Figure 3). The key operators along the value chain and their roles with PhP and generating packaging waste were first identified based on Holopainen's research [25]. This information was then elaborated during visits to individual operators and discussions with their experts. Figure 3 describes the ideal situation for packaging waste and pharmaceutical waste flows and, thus, excluded uncontrolled leakages to the environment, which happen occasionally.

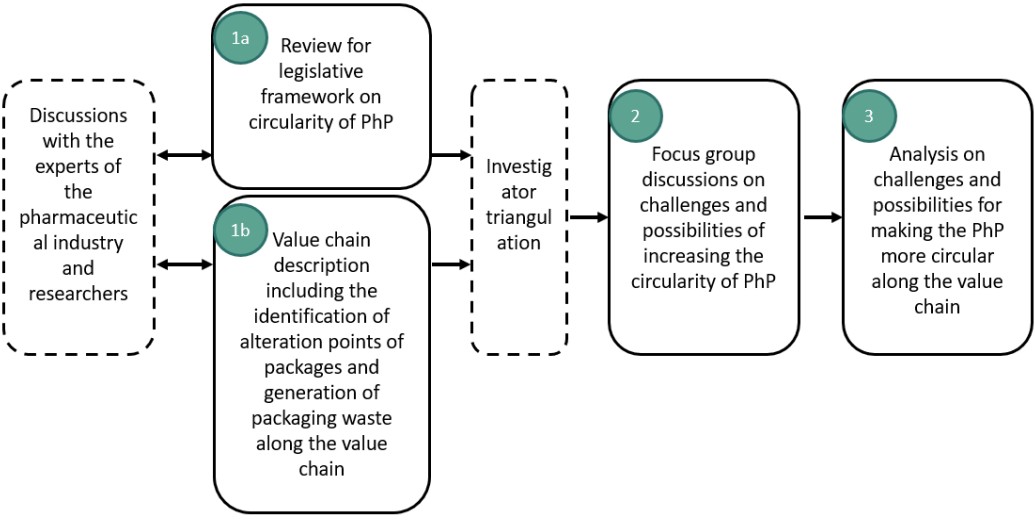

**Figure 1.** Phases of the study.

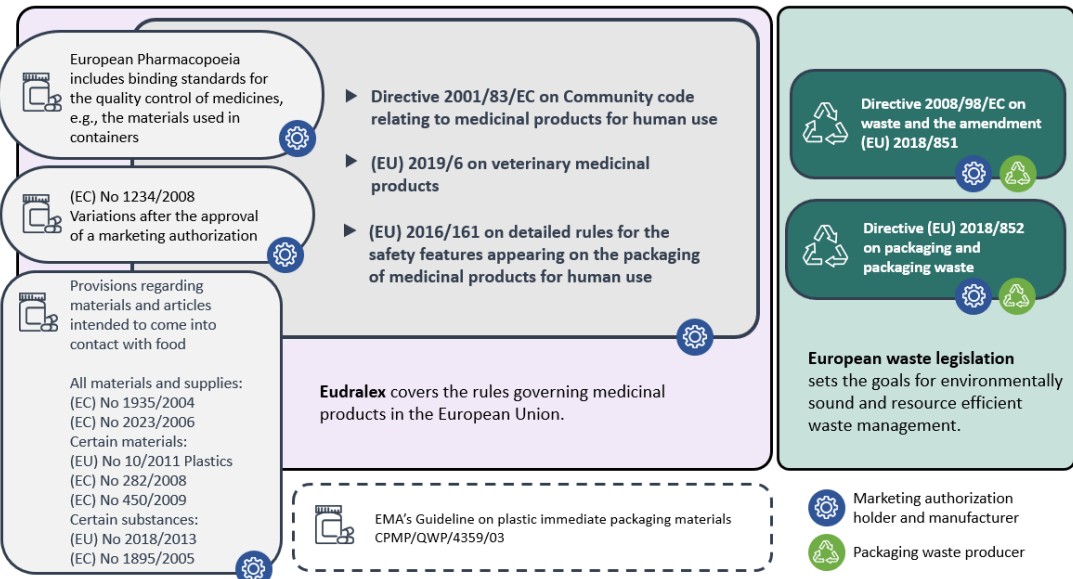

**Figure 2.** Legislative framework for PhP adapted from Honkanen [30].

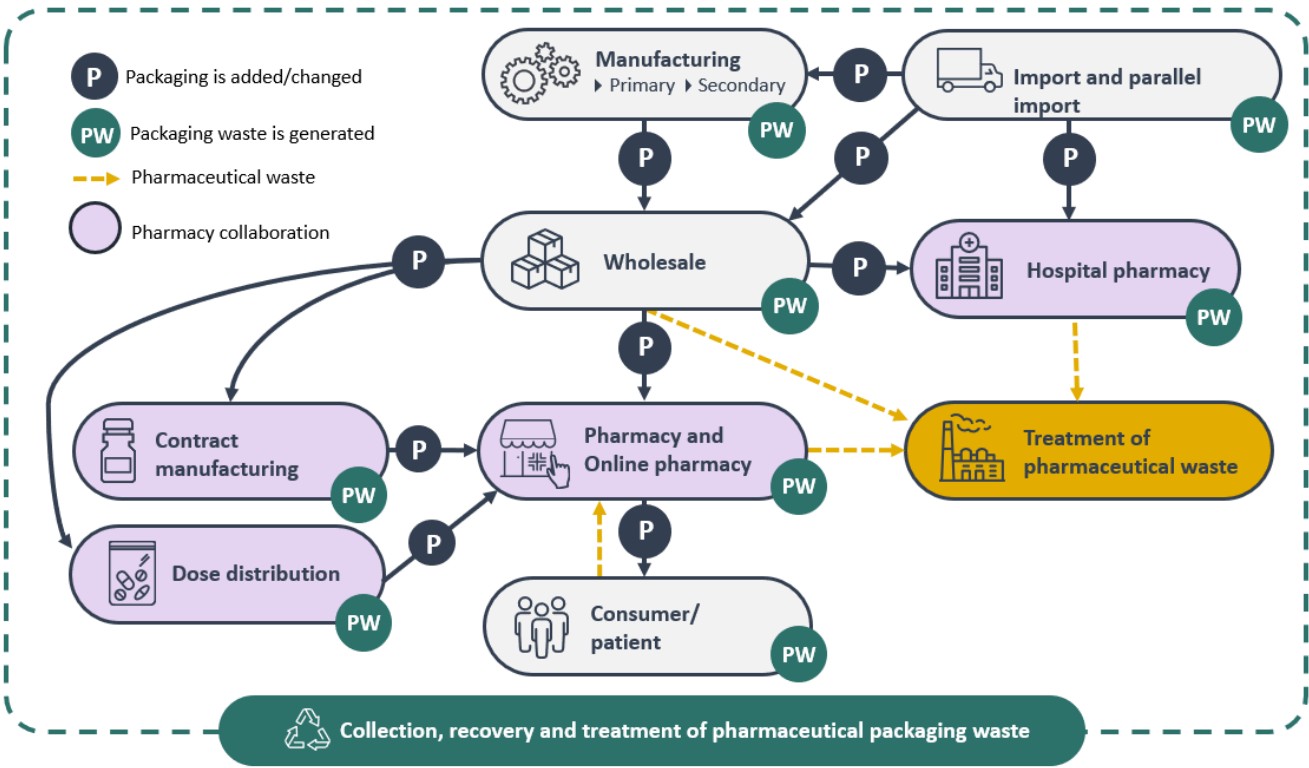

**Figure 3.** PhP Value chain adapted from Holopainen [25].

In the second phase, we focused on the other factors affecting circularity. By organizing a focus group discussion, we gathered data on perceived and experienced obstacles, challenges, hopes and possibilities for circularity of PhP along the value chain. The focus group discussion is a suitable method for qualitative research, as it aims to understand the studied phenomenon by gathering different perceptions and opinions [31]. In the discussions, the interaction between participants influences their answers and contributions, while the facilitator stimulates discussion with comments or subjects [32].

To reduce the researcher bias and increase the validity of the study, we used an investigator triangulation [33] in planning the focused discussions and the data collection. We also discussed and analysed the results by the group of researchers to gain a common understanding of the scope of the focus groups and to prevent inconsistent interpretations. Altogether, 24 representatives of key operators, authorities and experts from different parts of the defined value chain of pharmaceutical packaging (e.g., pharmaceutical industry, import, wholesale, dose distribution, pharmacy and waste management) participated in the discussions. The selection of participants was stratified by value chain phase. We identified key experts with the help of the stakeholder network of the SUDDEN project and the snowball sampling method [34], after which they were personally invited to discussions. The participants represented the best expertise on developing PhP circularity in their business in Finland. List of participants is presented in Appendix A (Table A1). We divided the participants into four discussion groups so that all groups covered the value chain operators as widely as possible. Groups were facilitated by the research team (one in each group) with the help of a semi-structured discussion framework. Each group discussed the same themes; (1) the challenges in developing PhP from a circular economy perspective, (2) improving circularity of PhP, (3) the key actors in increasing the sustainability of PhP and (4) what were the things that the participants could do. Each discussion lasted one hour. After four simultaneous group discussions, the results were discussed in a joint debate. Notes from the discussions were transcribed.

Finally, we analysed the results of the group discussions by categorizing them according to the value chain (Figure 3) phases to build an overall picture of the barriers, but also

the appropriate changes for making the PhP more circular. Further, the data concerning the barriers were thematically analysed [35] in a theory-based manner (deductively) according to the value chain phases, but also according to key factors such as institutional, economic, technological and sociocultural factors, adapted from, e.g., [7–10,36], regarding enabling but also hampering the circularity issues. Key concepts in the coding were regulations, guidelines, incentives, resources, investments, markets, profitability, information, material technology, recycling, digitalization, practices, co-operation and dialog, attitudes and awareness.

## 3. Results

### 3.1. The Legislative Framework

3.1.1. Legislation on Pharmaceutical Packaging

Pharmaceutical packaging is regulated strictly to ensure the safety of the pharmaceutical products for humans and animals and to avoid counterfeit medicines. As this case study applied to Finland, an EU member country, it was necessary to review EU legislation as well as some global recommendations and guidelines provided by the World Health Organization (WHO). National regulations were not specifically observed in this study, except for a few practical examples raised in the focus group discussions. The overview of the relevant legislation is summarized in Figure 2.

The basis for the PhP legislation was in the directive 2001/83/EC on medicinal products for human use, in the regulation (EU) 2019/6 on veterinary medicinal products and in regulation (EC) 1234/2008 on the examination of variations to the terms of marketing authorizations for medicinal products for human use and veterinary medicinal products. The foreword of the 2001/82/EC pointed out the necessity of binding regulations but also linked to the need for proper packaging.

> *'It is necessary to exercise control over the entire chain of distribution of medicinal products, from their manufacture or import into the community through to supply to the public, so as to guarantee that such products are stored, transported and handled in suitable conditions.'*

Directive 2001/83/EC and regulation (EU) 2019/6 described the content of the marketing authorization application and the information required on the packaging concerning the active pharmaceutical ingredients and the final product in storage, transport and use. They were also obliged to provide package information in the summary of product characteristics and give the instructions for labelling and package leaflet of the medicinal product. The safety features of the packaging of medicinal products for human use were also provided in (EU) 2016/161.

If any alterations, such as changes that may lead to the revision of the summary of product characteristics, labelling or package leaflet, are to be determined on the terms of marketing authorizations for medicinal products, they should be examined by a marketing authorization authority according to (EC) no. 1234/2008. This also applies to changes in PhP.

EudraLex is the collection of rules and regulations governing medicinal products in the European Union. In Volume 2 (for human use) and Volume 6 (for veterinary use), EudraLex presented the regulatory guidelines for the procedures for marketing authorization, but also for medicinal products, including the packaging information of medicinal products and classification for the supply, readability of the label and package leaflet requirements. Volumes 2C and 6C dealt with the presentation of the medicinal product covering pack size, design and composition. The guidelines emphasized that when presenting a range of pack sizes for a medicinal product, it is important that the principals of rational use of medicinal products are observed. The purpose of this guideline was to describe how the directive 2001/83/EC and regulation (EU) 2019/6 are implemented in the case of a marketing authorization. Even though other value chain operators would benefit from changes in the package design, size or material, it is always the manufacturer or the marketing authorization holder who has to apply for the new authorization.

The European Pharmacopoeia contains standardized specifications for the pharmaceutical industry, defining the quality of pharmaceutical preparations, their ingredients and containers [37,38]. Its binding standards provide a scientific basis for quality control during the entire life cycle of a medicinal product [37]. Additionally, a variety of packaging materials, such as blisters, plastics and articles with, for example, plastic layers that are in contact with the medicinal product and materials without a monograph in the European Pharmacopoeia or any other pharmacopoeia, is regulated by regulation (EU) 10/2011 on plastic materials and articles intended to come into contact with food. General regulations for all materials and supplies, but also other material-specific regulations such as EU 10/2011 and EU 282/2008 on recycled plastic materials and articles, cover a wide range of packaging. The use of recycled materials is not approved for raw materials [39]. The European Medicines Agency (EMA) published a guideline (CPMP/QWP/4359/03) that only applies to plastic primary packaging materials, but the approach given can be applied to other materials as well. The guideline describes the minimum requirements for the information to be provided in the marketing authorization.

Finally, the WHO published guidelines on packaging for pharmaceutical products. Environmental impacts are expressed, but also requirements of the other aspects of packaging, such as the protection of the quality of medicine. The WHO emphasized the need to reduce the volume and weight of packaging material and to ensure that environmentally friendly packaging is recyclable or degradable. They also highlighted that primary packaging, e.g., materials that have been in contact with toxic drugs, requires incineration, whereas uncontaminated packaging can be recycled [1].

### 3.1.2. Waste Legislation on Packages

Environmental legislation tackles packaging and packaging waste in the waste framework directive 2008/98/EC and the amendment (EU) 2018/851. The WFD set the principals of the waste hierarchy that should be applied in the waste legislation and waste policy of member states. According to the waste hierarchy, the most preferred option is to prevent waste, then reuse and recycle and the least preferred is disposal. Waste prevention means measures taken before a substance, material or product has become waste, reducing the quantity of waste, the adverse impacts of the generated waste on the environment and human health or the content of harmful substances in materials and products. The new amendment of the waste directive set ambitious goals for waste recycling that were tied also to the EU circular economy policy. The EU also introduced the extended producer responsibility (EPR) system to solve the waste management challenges of various product groups, packaging, etc. The EPR system follows two principals: the shift of responsibility upstream toward the producer and the provision of incentives to producers to include environmental elements in the design of their products [40], although producers' product design responsibilities are currently not very detailed. Therefore, the EPR system binds waste issues to the product's design and production phase (Figure 2). There are new binding targets for packaging recycling set in directive (EU) 2018/852 on packaging and packaging waste. According to it, 55% of plastic packaging must be recycled by the year 2030. Additionally, in 2018, the European Commission proposed a plastics strategy with an aim that by 2030, all plastic packaging items should be reusable or recyclable.

### 3.2. The Pharmaceutical Packaging Value Chain

In this study, the value chain of pharmaceutical packaging demonstrated the life cycle of medicinal products in Finland from imports through the manufacturing of pharmaceuticals, wholesale, dose distribution, contract manufacturing and pharmacy, all the way to the consumer (Figure 3). The manufacturing included also packaging design.

In the value chain of PhP, imports include both pharmaceuticals' raw materials, active medicinal ingredients and medicinal products, all of them packaged to secure the products. Especially the so-called parallel import, which refers to the import of a non-counterfeit product from another country without the license of the intellectual property owner, creates

packaging waste, since the secondary packaging of imported pharmaceuticals needs to be changed to fulfil the requirements set by the national authorities concerning packaging, packaging leaflets and labels in the two official languages, Finnish and Swedish. Companies utilise parallel import to benefit from the price variations within the European Economic Area. Pharmaceuticals are purchased with lower prices, which introduces competition and allows for the introduction of products with lower prices for consumers. In Finland, 34 companies have a license to operate as a pharmaceutical manufacturer [41] and imported products play an important role in the Finnish pharmaceuticals market [25]. Contract manufacturing covers the preparation of certain medicinal products in pharmacies.

Wholesale is a key operator in the PhP value chain, since wholesale receives pharmaceuticals and medicinal products from manufacturers and importers and distributes them further to pharmacies, dose distribution and contract manufacturers. Packaging waste is generated when the wholesale operator repacks PhPs according to orders and adds tertiary packaging to facilitate transportation. Reusable boxes are used in deliveries to pharmacies when possible. However, large orders need to be packed, for example, in cardboard boxes or on pallets using plastic film to secure the load to the pallet.

Contract manufacturing means that some pharmacies produce certain medicinal products by themselves or order them from a specific contract manufacturing pharmacy. A pharmacy's responsibility is to provide its customers with the medicines they need. The so-called ex tempore products are sometimes prescribed to patients in special situations where no industrially manufactured substituting medicinal products can be found for them. Contract manufacturing pharmacies pack the medicinal products again.

Dose distribution represents the extended pharmacy's contract manufacturing. Dose distribution serves pharmacies by dividing patients or customers' medicines into prescribed doses to help the customer take the medicines required. Dose distribution is carried out in a pharmacy or hospital pharmacy with the permission of a national authority [42] and it presents an essential part of pharmacy contract manufacturing. Packaging waste is generated in the activity, since the dose distributor receives the pharmaceuticals and medicinal products in packages of various sizes, empties the packages (primary and secondary) and distributes the medicines into small plastic bags. Further, medicines from hospital pharmacies are distributed to patients by health care personnel.

Pharmacies sell and distribute pharmaceuticals and medicinal products also to consumers. The tertiary packaging used during the transport of these products from wholesale to pharmacies generates packaging waste. Consumers utilize the medicines, and while doing so generate packaging waste, including both primary and secondary packaging. Old and unused pharmaceuticals create medical waste, which consumers return to pharmacies. Occasionally pharmacies return unused pharmaceuticals to wholesale operators, e.g., due to product error or removal of pharmaceuticals from the market. Both pharmacies and wholesale operators deliver the medical waste to hazardous waste treatment and PhP waste to appropriate waste collection. In Finland, all medical waste is classified as hazardous waste and pharmacies have determined an agreement with the Ministry of the Environment to organise the take-back system of medical waste from consumers. Along with medical waste, consumers tend to also return pharmaceutical packaging waste to pharmacies, although they are instructed not to. From a global perspective, Finland's system for collecting pharmaceutical waste works well, as it is comprehensive and waste management is appropriate [43].

### 3.3. Focus Group Discussions

The results from the focus group discussions are presented below according to specific challenges and possibilities to promote the circularity of PhP by various value chain phases. The thematically analysed results are summarized in Table 1.

**Table 1.** Barriers and possibilities according to value chain phase.

| Value Chain Phase | Barriers | Possibilities |
|---|---|---|
| Manufacturing | R: Regulations on pharmaceuticals make changes in packaging difficult and slow. R: Lack of incentives to develop PhPs from environmental perspective. R: National requirements for medicine packaging might be stricter compared to European ones. Need for uniformity. T: Knowledge gap in comparison with different packaging materials and their environmental impacts in the different stages of the value chain. T: Lack of recommendations for recycling instructions on the package and information sheet. T: Need for tertiary packaging to seal the package to increase the safety and decrease the use of falsified medicines. T: Maintaining the cold chain of pharmaceuticals and protecting the medicines from freezing can lead to overpacking. E: Customers do not want to keep large stocks, so the number of deliveries and shipments is increasing. S: Lack of dialog between waste management and packaging developers. | Manufacturers need to be challenged to offer circular packaging solutions. Economic support should be allocated for experiments, and authorities should allow testing and piloting. Authorities should join the dialog on the development of the pharmaceutical packaging. Packaging design guidelines from the perspective of circularity. A new electronic packaging leaflet. Implementation of consistent EU pictogram system for all packaging. Use of monomaterials in plastic packaging enhances circularity. Film-coated tablets could reduce drug residues in the packaging. The efficiency of transport system of pharmaceuticals should be improved, and LCA-based knowledge is needed. Instructions also to manufacturers and wholesale on how packaging should be recycled. |
| Wholesale | E: Amount of obsolete medicine waste from wholesale for disposal means also packaging waste. E: Due to competitive reasons, all packaging sizes cannot be kept on the market. E: Finland is a small market area. T: Keeping up the cold chain of pharmaceuticals and protecting the medicines from freezing can lead to overpacking. | Because of the large volumes of PhPs, wholesale is in a key role when developing possibilities to recycle the packaging. Provide instructions also to manufacturers and wholesale on how packaging should be recycled. The efficiency of the transport system of pharmaceuticals should be improved. |
| Dose distribution | R: Regulations hinder the possibility of dose distribution to order larger packs of medicines abroad. T: Keeping up the cold chain of pharmaceuticals and protecting the medicines from freezing can lead to overpacking. | Dose distribution can be developed to use larger packs, which reduces blister amounts. The efficiency of transport system of pharmaceuticals should be improved. |
| Pharmacy and hospital pharmacy | R: The dose delivery packs are not allowed for hospital use because of marketing authorizations. R, E: Delivery obligation restricts the pack sizes available. T, E: Same size packaging is user-friendly for medicine storage and delivery. S: Lack of dialog between waste management and pharmacies. | Pharmacies could follow the amounts of returned PhP by consumers. |

**Table 1.** *Cont.*

| Value Chain Phase | Barriers | Possibilities |
| --- | --- | --- |
| Consumer | R, E: Restrictions in reimbursements for medicine expenses have an impact on what package size customer chooses.<br>E: Bigger packages can be cheaper for customers.<br>T: Consumers do not necessarily understand the packaging waste sorting instructions.<br>S: Commitment to medication. The medicines are not necessarily used according to the doctor's advice which creates unnecessary wastes. | Information campaigns are needed. EU pictograms would ease sorting. Consumer research should be carried out on how much medication is left unused and for what reason. |
| Waste management | T: Lack of information on whether the pre-treatment of plastic packages is enough to provide safe cycles.<br>T: Lack of data on PhP waste volumes.<br>T: Recycling of pharmaceutical packages is hindered by certain materials or packaging types, e.g., PVC or multilayer packaging.<br>S: Lack of dialog between waste management and pharmacies. | Chemical recycling is expected to ease the plastic packaging recycling. |

### 3.3.1. Challenges in the Development of Pharmaceutical Packaging from a Circular Economy Perspective

Changing pharmaceutical packaging is very challenging due to the existing regulations and marketing permits. Participants agreed that the development of packaging is strongly tied to the requirements of medicines. However, binding targets for the recycling of plastic packages have been set and lists of banned materials are being considered at the European level. Additionally, the producer responsibility fee set in waste legislation is going to be staggered according to recyclability soon. All of this is likely to put pressure on packaging manufacturers to develop packaging also from the perspective of the circular economy. However, it has been stated that more incentives are needed for the development of new packaging materials. The change is perceived to be slow.

Stakeholders expect that new materials are likely to be developed to replace plastics. It was stated that the recycling of PhP is hindered by certain material choices or packaging types, e.g., polyvinyl chloride or multilayer packaging. However, participants agreed that packaging materials are selected according to the requirements of medicines, since the protection of pharmaceuticals is the key requirement due to patient safety. It was also questioned if changing or reducing materials would necessarily bring the best performance in circularity. For example, more monomer materials may be needed for packaging to compensate for multilayer materials which eventually lead to an increase in the amount of waste. Therefore, it was pointed out that it is necessary to focus on reducing the loss in the whole supply chain. By following the current care guidelines and rational pharmacotherapy, packaging waste can be reduced.

The experienced rigidity of the PhP system due to regulatory restrictions and administrational practices affects several phases of the value chain, e.g., the larger dose delivery packs are not allowed for hospital use as dose delivery packaging has a marketing authorization for this purpose only. However, hospital pharmacies would presently benefit from larger packaging sizes due to the similarities with dose distribution operators. Ordering bulk packaging directly from the manufacturer, even from abroad, for dose distribution would represent a circular solution, but it is currently not possible. Furthermore, due to competitive reasons, all packaging sizes cannot justifiably be kept on the market. Addi-

tionally, national restrictions on reimbursements for medicine expenses have an impact on the package size a pharmacy customer chooses. A bigger package can also be cheaper for customers than a smaller one. Administrational practices vary from country to country.

The possibility of dose distribution to order larger package sizes of medicines from abroad is currently not possible in Finland, since the product size should have a marketing authorization in both importing and exporting countries. It was observed that the introduction of this 'foreign package' option in Finland could lead to a reduction in PhP waste. In the system, any concentrations and pack sizes would have a marketing authorization in Finland and the pharmaceutical company could import a suitable packaging size for dose distribution. This foreign package system is currently used in Sweden and Norway, whereas the dose distribution is practiced only in Nordic countries and a few other European countries.

Consistent recommendations from the authorities were called for on how companies themselves could add recycling instructions on the packaging and information sheets. It was also hoped that there would be no need for fees for these changes. A pharmacy representative stated that consumers do not necessarily understand the sorting instructions when returning used medicines to the pharmacy. In the pharmacy collection point, all packaging is often mixed up, including the ones that are not advised to be brought for collection at all, such as secondary packaging. Therefore, there is a need for uniform symbols on the packaging.

Potential pharmaceutical residues in primary packages were considered to be a threat. For example, it is not known whether the packages can be safely recycled. Certain pharmaceuticals can be retained inside the plastic materials used in primary packages, depending on their chemical properties. It was suggested that some pharmaceuticals may also leave residues on the surfaces of these primary packages, which can easily be washed away. Pretreatments in mechanical plastic recycling, e.g., washing processes of the plastic waste, may create the threat of an uncontrolled leakage of pharmaceutical residues to water systems because there is no suitable wastewater treatment technology to remove pharmaceuticals from effluents. A relief from chemical recycling is expected for plastic recycling, but the pre-sorting of packaging waste is still needed.

Several distinct challenges for the sustainability of packaging and recycling arose during the discussion. For example, Finland's climate brings an additional challenge to the logistics of medicines, but also their packaging. Medicines require extra packing to protect them from freezing during the winter. On the other hand, in summer, the cold chain of medicinal products must not be broken. It was noted that monomaterial aluminium blisters are not well suited for dose distribution, even though they would be easy to recycle.

### 3.3.2. Possible Changes in Pharmaceutical Packaging

Participants recognised that there are measures that value chain operators can carry out independently. Manufacturers of pharmaceuticals and packaging designers need to be challenged to offer new packaging solutions. Economic support should be allocated for experiments and authorities should allow testing and piloting. It was hoped that authorities would join the dialog between stakeholders on the development of the PhP as it was questioned that Finnish requirements for medicine packaging might be stricter compared to those in Europe.

There is a need for more comprehensive research on several issues. For example, information is needed on the types of packaging, e.g., quantities, types, materials and structure, used by the pharmaceutical industry in both primary and secondary packages. Additionally, data on PhP waste collected in pharmacies should be gathered. It was stated that a deeper understanding about environmental and economic benefits and effects is required if the packaging material is changed. In addition, there should be research on the environmental impacts in the different stages of the value chain to identify the most focal points.

Ongoing cooperation with waste operators, pharmacies and pharmaceutical industry is necessary. It was suggested that there could be comprehensive packaging design guidelines from the perspective of circularity. Even though some general instructions for package design already exist, no specific criteria exist for PhP. There was a consensus that the current strategy of using package data sheets should be abolished. A new electronic packaging leaflet would be technically simple to implement, but this needs regulatory changes at the EU level. An electronic packaging leaflet would decrease the amount of secondary packaging waste, especially in a situation when changes in the leaflet are needed and already-printed leaflets with outdated information immediately become waste. Guidelines for the sorting and disposal of medicine packaging are needed and the use of consistent EU pictograms for all packaging could be a solution for sorting packaging waste. Pictograms have already been introduced in Denmark, Sweden and Norway, and are widely being discussed in the EU. Customer awareness and practices require specific attention and information campaigns for customers were suggested.

The pharmaceutical operators, such as manufacturers and wholesale, should be advised on how to sort the PhP waste generated in their operations, as it was noted that even small amounts of "wrong materials" can cause problems in recycling. There was also a discussion on whether film-coated tablets could reduce the drug residues in the packaging and, thus, make recycling safe. The wholesale of medicines is a significant operator along the value chain as huge amounts of packaging material, e.g., carton and plastic, are dealt with and wasted there. It was highlighted that wholesale is in a key role when developing recycling options. It was also suggested that the efficiency of the transport system of pharmaceuticals should be improved.

### 3.3.3. Categorized Barriers and Possibilities

The barriers were categorized into regulatory (R), technological and informational (T), socio-cultural (S) and economic (E) aspects in Table 1. A wide range of barriers exists in all value chain phases. Many regulatory but also technological and informational barriers hamper the development of PhP. In some cases, barriers along the value chain are interconnected. For example, regulations may lead to practices which are the most economical but not necessarily the most sustainable. Additionally, in the waste management phase, the barriers to recycling are connected to production and the design phase.

Even though the production phase is loaded with various barriers to circularity, there are also possibilities. Stakeholders identified many potential measures which do not even require any changes in legislation. Many of the actions are connected to the need of dialog between value chain stakeholders and other informational improvements. The chemical recycling of PhP is hoped to solve many problems linked to clean material cycles.

## 4. Discussion

This study showed that stakeholders perceive that the value chain is in need of adaptation to the growing requirements for sustainability. They also identified a variety of regulatory, economic, technical and socio-cultural barriers in developing PhP circularity through all stages of the value chain. Out of all the phases of the value chain, the manufacturing of medicines was the key phase when increasing the circularity of PhP, although it is possible to develop circularity at all stages of the value chain. Legislation on pharmaceuticals sets the rules and framework for developing PhP. The requirements of legislation for packaging from the health and safety perspective are unquestionably primary in relation to requirements for the environment. In this study, packaging technology did not appear as a major barrier in the development and manufacturing phase, whereas the lack of data on environmental impacts of packaging materials was identified as a key obstacle. Correspondingly, Lorenzini et al. [22] recognised that technology and legislation are stronger drivers for the development of PhP than sustainability. Consumers are in a key position, as they have a strong will to recycle packaging [6]. Our results showed that socio-cultural barriers play a role in increasing the circularity. Dialogue between key actors in the value

chain is needed. For example, the pharmaceutical authorities should join the discussions on environmental aspects. Additionally, waste operators should contribute with their knowledge about recyclability. The lack of interaction was also noticed by Rizos et al. [36], who stated that especially small and medium-sized enterprises are lacking support in their supply and demand network.

To promote the circularity of PhP in the manufacturing phase, the value chain actors identified a large variety of possibilities, such as cross-sectoral expertise, economic support for piloting in package design, package design guidelines, digitalisation or LCA-based knowledge on circular solutions. As the development of PhP materials is ongoing [23], it was surprising that the stakeholders paid only a small amount of attention to optional packaging materials. Only monomaterials, which could ease the recycling of problematic blister packaging, were discussed. Oliveira et al. [39] studied the sustainability of blisters and found that LCA studies were still needed in order to reduce materials or establish changes to materials for blisters. However, principles for circular packaging design exist and the development of PhP materials has correlations with food packaging [44], as both have specialized requirements in terms of the purity or quality of material and focus on protecting the content from microbiological, physical and chemical disturbances. In addition to packaging design, we found that packaging leaflets should include additional information on the environmental impact of PhP and its proper disposal. Electronic leaflets and a uniform pictogram system for waste sorting in the packaging is singled out as a potential solution to increase recycling. Pictograms are already used widely in the Nordic countries. Additionally, a harmonized system in EU is being discussed and interest in the pictograms is rising also globally [45].

The legislative review (Figure 2) showed that from the value chain perspective, EU regulations on PhP are mostly tied to the manufacturing and marketing phase. Strict marketing authorization practices are restricting factors in the development of packaging materials. Even small changes in the packaging material can cause a significant administrational burden for manufacturers. The eco-design of products is loaded with expectations for enhancing the sustainability and the circular economy, e.g., [46,47]. However, PhP design faces strict regulations, incomplete information about suitable packaging materials, but also rigid and established practices along the value chain, making it difficult to change packages. Waste regulations deal mainly with package waste management. The EPR system has, thus far, increased the amount of recycled packaging, but has not had much impact on packaging design [40,48]. Yet, the EPR legislation has some means to encourage the producers for eco-design (e.g., the modulated fees), but stakeholders perceived that there could be even stronger incentives.

The value chain examination illustrated (Figure 3) that the distribution of pharmaceuticals along the value chain increased packaging and packaging waste amounts. Additionally, the practices of intermediate operators such as dose distribution, wholesale, storage and transportation were recognized as phases where material losses occur and substantial amounts of PhP waste are generated. Many of the obstacles to circularity identified in this study were interconnected. Regulations lead to certain practices, which can quite often be cost-efficient but not material-efficient since they may lead to the generation of excess amounts of waste. For example, it is laborious and costly to change the marketing authorization for a particular medicine package size and drug. Furthermore, pharmacies do not always have all pack sizes in storage, due to the possible obsolescence of medicines. Therefore, customers may be forced to buy medical product in a pack size that can create additional waste later. We believe that existing, old procedures which have developed over the years and have been adopted by operators need critical scrutiny. Yet, the environmental impacts for the whole lifecycle must be observed when determining changes in packaging [39].

There is a potential conflict between environmental risk and the promotion of material recycling that PhP faces as a special packaging group. The gap between ambitious recycling targets and existing recycling systems for complex package materials was also observed by

Soares et al. [49] in their study on the recycling of multi-material plastic packaging. The proportion of plastics as packaging materials is significant and their recycling has been recognized to be troublesome from the perspective of clean and safe cycles [1] (p. 142). This study revealed the worries of stakeholders that are connected to the recycling of plastic PhP. Chemical recycling solutions are expected to alleviate the above-mentioned contradictions. Chemical recycling has high potential for miscellaneous and contaminated plastic waste, where separation is neither economically viable or not completely technically feasible [50]. Since mechanical recycling mostly lacks the ability to remove hazardous substances during the recycling process [51], the development of chemical recycling solutions could be the way for a higher plastics recovery. However, even when using chemical recycling solutions, plastic waste must first be mechanically treated and it has been questioned if decomposing plastics into monomers is a feasible and environmentally wise solution [51]. In Europe, only one third of plastic waste is recycled [51,52], so all plastic waste streams are needed to recover to reach the ambitious recycling targets set by the EU.

The academic literature lacks studies on PhP and sustainability. We need more research on the environmental impacts connected to PhP materials and the performance of the PhP value chain. Data on the quantity and quality of PhP are urgently required for such research and for demonstrating the significance of material looping in the big picture of packaging. This study also highlighted the lack of data on waste volumes as a barrier to increasing circularity, as discussed by Salmenperä et al. [9]. However, when developing packaging, we should not forget that some environmental burdens connected to PhP may need to be tolerated to maintain the most important function of packaging, namely, protecting the medicines.

## 5. Conclusions

The pharmaceutical packaging sector is no doubt a very specific sector and needs extra attention in promoting circularity, as health and safety are the primary perspectives in decision making. Legislation, a lack of information or interaction between stakeholders and rigid practices were identified in our study as blocks for a more circular product design. Earlier studies revealed that similar barriers and drivers for promoting the circular economy were to be found in different sectors. Therefore, we could learn from previous circular economy-focused studies in comparable sectors, for example, in the food industry. The pharmaceutical packaging sector could follow the circularity decisions determined in the food packaging sector about the possible use of chemically recycled plastic materials that come into contact with food. In changing packaging practices to more circular ones, the causal links along the value chain need to be understood. Additionally, the environmental impacts of circular solutions must be studied. A change of mindset is needed for all activities in the value chain. This occurs most effectively by increasing interactions between stakeholders and by investing in the quality of environmental information.

**Author Contributions:** Conceptualization, H.S. and S.K.; methodology H.S. and S.K.; investigation, H.S., S.K., H.D. and P.F.; writing—original draft preparation, H.S.; writing—review and editing, H.S., S.K., H.D. and P.F. All authors have read and agreed to the published version of the manuscript.

**Funding:** This research was funded by the Academy of Finland's Strategic Research Council grant number 320233/Sustainable Drug Discovery and Development with End-of-Life Yield–SUDDEN.

**Informed Consent Statement:** Informed consent was obtained from all subjects involved in the study.

**Data Availability Statement:** Not applicable.

**Acknowledgments:** The authors would like to thank all the focus group participants, especially Outi Honkanen and Reijo Kärkkäinen for valuable comments.

**Conflicts of Interest:** The authors declare no conflict of interest.

## Appendix A

**Table A1.** List of Participants in Focus Group Discussions.

| Number of Participants in Groups | Organisation |
|---|---|
| Group 1 | |
| Participant 1#, | Pharmacy |
| Participant 2#, | Pharmaceutical company |
| Participant 3#, | Distributor of pharmaceuticals |
| Participant 4#, | University |
| Participant 5#, | Parallel import of medicines |
| Participant 6#, | Governmental research institute |
| Participant 7#, | University |
| Facilitator | |
| Group 2 | |
| Participant 8#, | Dose dispending |
| Participant 9#, | Research and communication |
| Participant 10#, | State administration |
| Participant 11#, | Plastic waste recycling company |
| Participant 12#, | Pharmaceutical information centre |
| Facilitator | |
| Group 3 | |
| Participant 13#, | Dose dispending |
| Participant 14#, | University |
| Participant, 15#, | Municipal waste management company |
| Participant 16#, | Distributor of pharmaceuticals |
| Participant 17#, | Pharmacy |
| Participant 18#, | State administration |
| Facilitator | |
| Group 4 | |
| Participant 19#, | Pharmaceutical company |
| Participant 20#, | Governmental research institute |
| Participant 21#, | Dose dispensing |
| Participant 22#, | Producer responsibility of packaging |
| Participant 23#, | Distribution of pharmaceuticals |
| Participant 24#, | University |
| Facilitator | |

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
