# Peer review of "Increasing the Circularity of Packaging along Pharmaceuticals Value Chain"

_sustainability, doi:10.3390/su14084715_

Round 1

Reviewer 1 Report

The manuscript looks interesting but it needs to be more concise and focus to certain topic. 
The abstract can be better written.  The current state of the art is not fully described in the introduction section. Analysis of the problem under investigation is not fully justified.. There are many redundant (.e.g Pharmaceutical packaging (PhP)), the objectives are confusing (. e.g, the authors mentioned: the study explore ....) then they mentioned: the aim of the study..... The method section should describe the steps of scientific work. it  looks like an introduction section and/or result and discussion.
Statistical analysis in not described, so it is hardly to decide., The flow of result section is not in accord with sequence of M&M.   
Reference    some times are cited as numerical order and alphabetic order (e.g ref 14)

Reviewer 2 Report

The manuscript focus on circular economy in the pharmaceutical waste packaging industry is of interest in the field considering the importance of waste recycling in the sector. The content of the manuscript is good but needs some improvement before it can be published.

  1. The aim of the manuscript should be more focused. Based on what is in the manuscript, it is focused on three directions as indicated in the last paragraph of the introduction.
  2.  In the methodology, the authors reviewed the legislations but did not provide the keywords they used in the review. 
  3. Focused group discussion was conducted with 25 participants, but the socio-demographic characteristics of the participants such has which section of the sector they wok in, their work experience, knowledge or awareness about circular economy was not indicated.
  4. The results can be presented better in terms of the percentage/ number of the participants contribute to a certain theme rather than stating that all participants agree to every question. 
  5. Although chemical recycling technology was proposed as a solution, the advantages and disadvantages in relation to the focus of the study must be highlighted which is absent in the manuscript.
  6. The conclusion needs to highlight the main contribution of the manuscript.
  7. The abstract did not include the aspect of looking further into the environmental aspect of the value show which seems like a contribution of the manuscript.

Reviewer 3 Report

The manuscript "Increasing packaging circularity along the pharmaceutical value chain" provides a valuable and original overview of the complex aspects of circularity in primary packaging that comes into contact with pharmaceuticals.

The manuscript describes current primary packaging solutions, mainly plastics and multilayer materials, and correctly identifies the limitations and specific aspects of mechanical or chemical recycling of these materials in direct contact with pharmaceuticals.

The major driving forces for promoting circularity in the pharmaceutical packaging value chain are also clearly identified and explained.
A simple application of recent European regulations and laws [for strict waste management and implementation of economic circularity in all industrial sectors] is not evident in the pharmaceutical sector, where health and safety aspects must remain a priority in decision-making. 

One of the strengths of the manuscript is the very good knowledge of the pharma value chain, which has enabled the identification of the main existing barriers and some solutions to address them so far. Another strong point of this manuscript is the very good understanding of the interconnected factors playing in the pharma packaging design and material choses. 

The cited references are well-chosen and cover the entire range of aspects involved in the packaging circularity along the pharmaceutical value chain. Despite a special focus on the Finish situation, the manuscript offers a comprehensive overview of the circularity issues, transposable to the entire European pharmaceutical sector.

In conclusion, the manuscript is very clear and well-structured, cites very recent works in the field and provides a pertinent view concerning the feasibility and possibilities of implementing sustainability and circularity objectives in the pharmaceutical sector.

I recommend this manuscript for publication as it is.

Author Response

We value your opinion concerning our manuscript. Thank you.

Reviewer 4 Report

Dear authors,

There are some points where the paper could be improved. Please consider the following suggestions.

  1. We suggest a literature review regarding CE and approaches to promote it, to be included in the introduction. This would help the reader to understand better the significance of your work.

The following could be considered:

Nikolaou IE, Tsagarakis KP (2021) An introduction to circular economy and sustainability: Some existing lessons and future directions. Sustain Prod Consum 28:600–609. https://doi.org/10.1016/j.spc.2021.06.017

Zorpas, A.A.; Doula, M.K.; Jeguirim, M. Waste Strategies Development in the Framework of Circular Economy. Sustainability 2021, 13, 13467. https://doi.org/10.3390/su132313467

Karayılan, S., Yılmaz, Ö., Uysal, Ç., & Naneci, S. (2021). Prospective evaluation of circular economy practices within plastic packaging value chain through optimization of life cycle impacts and circularity. Resources, Conservation and Recycling, 173, 105691.

Meherishi, L., Narayana, S. A., & Ranjani, K. S. (2019). Sustainable packaging for supply chain management in the circular economy: A review. Journal of cleaner production, 237, 117582.

2. The scientific background of the used methodology is not presented. We suggest to be explained and some relevant references to be included. This would help the reader to understand better the scientific soundness of your work.

  1. Figures include spell check correction red underlines. Please remove them.

Round 2

Reviewer 1 Report

No additional. Comments

Reviewer 2 Report

The author(s) have addressed all my comments and the manuscripts have been improved significantly. The manuscript however need to undergo a  minor spelling check before it can be published.